# MoRAgent: Parameter Efficient Agent Tuning with Mixture-of-Roles

**Jing Han** [* 1]   **Binwei Yan** [* 2]   **Tianyu Guo** [2]   **Zheyuan Bai** [2]   **Mengyu Zheng** [2]   **Hanting Chen** [2]   **Ying Nie** [† 2]

## Abstract

Despite recent advancements of fine-tuning large language models (LLMs) to facilitate agent tasks, parameter-efficient fine-tuning (PEFT) methodologies for agent remain largely unexplored. In this paper, we introduce three key strategies for PEFT in agent tasks: 1) Inspired by the increasingly dominant *Reason+Action* paradigm, we first decompose the capabilities necessary for the agent tasks into three distinct roles: reasoner, executor, and summarizer. The reasoner is responsible for comprehending the user's query and determining the next role based on the execution trajectory. The executor is tasked with identifying the appropriate functions and parameters to invoke. The summarizer conveys the distilled information from conversations back to the user. 2) We then propose the Mixture-of-Roles (MoR) framework, which comprises three specialized Low-Rank Adaptation (LoRA) groups, each designated to fulfill a distinct role. By focusing on their respective specialized capabilities and engaging in collaborative interactions, these LoRAs collectively accomplish the agent task. 3) To effectively fine-tune the framework, we develop a multi-role data generation pipeline based on publicly available datasets, incorporating role-specific content completion and reliability verification. We conduct extensive experiments and thorough ablation studies on various LLMs and agent benchmarks, demonstrating the effectiveness of the proposed method. This project is publicly available at https://mor-agent.github.io/.

## 1. Introduction

Large language models (LLMs), trained on extensive corpora, have exhibited impressive performance across a wide range of natural language processing (NLP) tasks. Furthermore, LLMs have also demonstrated their capability in more challenging tasks, such as function-calling as AI agents. For instance, certain research endeavors (Hong et al., 2023; Shen et al., 2024b; Wang et al., 2024) employ prompt engineering methodologies. This approach involves the design of meticulously crafted prompts to effectively activate the agent-related capabilities in LLMs such as reasoning and tool-utilization. These works have shown superior agent performance in large commercial models, like ChatGPT and GPT-4. For open source LLMs (Touvron et al., 2023; Yang et al., 2024), although they have achieved considerable success across various NLP tasks, they still lag behind commercial models when acting as agents. Consequently, another approach has emerged: fine-tuning models using agent-specific data, which has yielded remarkable results on a variety of agent tasks (Qin et al., 2023; Zeng et al., 2023; Chen et al., 2024).

Most existing research on agent fine-tuning focuses on full-parameter fine-tuning, which presents two significant challenges. First, the computational resources required to handle billions or tens of billions of model parameters pose a substantial barrier to the widespread adoption of agents. Second, full-parameter fine-tuning compromises the general capabilities of the original base model, thereby limiting the flexibility of users to seamlessly switch between general tasks and agent-specific tasks. To address these challenges, Parameter-Efficient Fine-Tuning (PEFT) methods offer a viable solution (Houlsby et al., 2019; Li & Liang, 2021; Lester et al., 2021; Hu et al., 2021). Among these, Low-Rank Adaptation (LoRA) (Hu et al., 2021) is a prominent method that introduces low-rank adaptation matrices to simulate gradient updates while keeping the pre-trained model weights frozen. LoRA achieves the performance comparable to full-parameter fine-tuning across a range of downstream tasks while requiring significantly fewer computational resources.

However, directly applying LoRA for fine-tuning agents often yields performance that is significantly inferior to that achieved through full-parameter fine-tuning. Successfully accomplishing agent tasks typically requires LLMs to simultaneously exhibit multiple capabilities. For instance, LLMs must first comprehend the user's query and perform a reasonable analysis and planning, demonstrating the ability of reasoning. Subsequently, it needs to invoke the correct

---

[*]Equal contribution  [1]School of Artificial Intelligence, Beijing University of Posts and Telecommunications [2]Huawei Noah's Ark Lab. [†]Correspondence to: Ying Nie <ying.nie@huawei.com>.

*Proceedings of the $42^{nd}$ International Conference on Machine Learning*, Vancouver, Canada. PMLR 267, 2025. Copyright 2025 by the author(s).

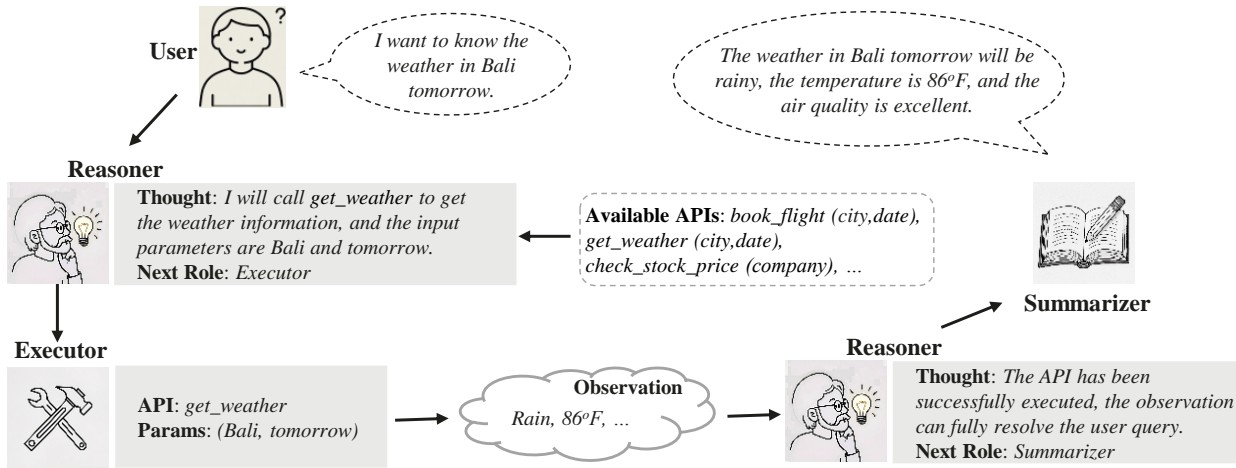

Figure 1. Workflow example of multiple roles collaborate to accomplish one agent task.

functions with suitable parameters, reflecting the ability of execution. After multiple rounds of interaction with the real environment, it should organize the conversation history and provide feedback to the user, exhibiting the ability of summarization. Learning multiple capabilities simultaneously is challenging for the parameter matrix, particularly when constrained to a low-rank form.

In this paper, we first decompose the capabilities necessary for the agent into three distinct roles: reasoner, executor, and summarizer. Specifically, the reasoner is responsible for comprehending the user's query and determining the next role based on the execution trajectory. The executor is tasked with identifying the appropriate functions and parameters to invoke, informed by the analysis of reasoner. The summarizer organizes the conversations history and conveys the distilled information back to the user. The pipeline of the multi-roles is illustrated in Figure 1. It should be noted that the summarizer is engaged to provide user feedback only when the reasoner deems the user's query fully addressed or, after multiple invocations of the executor, still cannot resolve the user's query and decides to give up.

To better characterize these capabilities, we then propose the Mixture-of-Roles (MoR) architecture, which comprises three specialized LoRA groups, each designated to fulfill a distinct role. By focusing on their respective specialized capabilities and engaging in collaborative interactions, these groups collectively accomplish the overall agent task. In practice, each group consists of a different number of LoRA modules. Our guiding principle is to allocate more LoRAs to relatively important roles and fewer to less important ones, aiming to achieve better performance with fewer trainable parameters. Additionally, we introduce both a rule-based role-aware gate and learnable token-aware routers to more reasonably allocate LoRAs to the input features. During the training process, auxiliary balance loss and orthogo-

nal loss between LoRAs are further introduced for better optimization.

To effectively fine-tune the framework, we develop a multi-role data generation pipeline based on publicly available datasets, incorporating role-specific content completion and reliability verification. Specifically, we prompt GPT4o to fill in missing multi-role content in agent data, such as thought and summary. The completed execution trajectories are further evaluated for quality by DeepSeek-V3 (Liu et al., 2024a), with low-quality samples being filtered out. Subsequently, we unify the data format to facilitate downstream fine-tuning. In practice, we observe certain issues in the outputs of executors within some samples, such as selecting functions outside the candidate list, encountering runtime errors, or failing to resolve user problems despite error-free execution. To address these issues, we adopt a hybrid approach combining manual corrections and LLM to further enhance its reliability.

We extensively evaluate the proposed method on various benchmarks, including StableToolBench (Guo et al., 2024), BFCL (Yan et al., 2024), GSM8K (Cobbe et al., 2021), and MATH (Hendrycks et al., 2021). For example, on Stable-ToolBench, our method achieves improvements in DFS pass rates of 40.6% and 14.2% for Llama3.2-1B-Instruct (Touvron et al., 2023) and Phi-3.5-mini-Instruct (Abdin et al., 2024), respectively, while introducing only an additional 0.16B and 0.36B trainable parameters.

## 2. Related Work

### 2.1. Agent Tuning

Fine-tuning open source LLMs on agent-specific data has shown to be an effective approach for developing agent capabilities. ToolLLM (Qin et al., 2023) presents a general

tool-use framework encompassing data construction, model training, and evaluation. The automatically constructed instruction-tuning dataset for tool invocation using Chat-GPT has been widely adopted. AgentTuning (Zeng et al., 2023) focuses on improving the agent capabilities of LLMs themselves without compromising their general abilities. To achieve this, they employ a hybrid instruction-tuning strategy by combining AgentInstruct with instructions from general domains. Gorilla (Patil et al., 2023) constructs the APIBench, a large corpus of APIs with complex and often overlapping functionality by scraping ML APIs from public model hubs. Toolformer (Schick et al., 2023) incorporates a range of tools, including a calculator, a Q&A system, a search engine, a translation system, and a calendar. KwaiAgents (Pan et al., 2023) and ModelScope-Agent (Li et al., 2023) presents a comprehensive framework spanning over tool-use data collection, tool retrieval, customized model training, and evaluation for practical real-world applications. Many works have also been proposed to synthesize higher quality data for agent tasks. APIGen (Liu et al., 2024d) introduces a function-calling data generation pipeline to facilitate the fine-tuning of function-calling LLMs by providing high-quality, diverse datasets that better reflect the variability and complexity of real-world API use. xLAM (Zhang et al., 2024) and ToolACE (Liu et al., 2024c) further discuss the data pipeline including data unification, augmentation, quality verification, general instruction data synthesis, and preference data generation.

In addition to a single model, multiple models working together to complete agent tasks has recently attracted the interest of many researchers. Mobile-Agent-v2 (Wang et al., 2024) presents a multi-agent architecture for mobile device operation assistance, which includes planning agent, decision agent, and reflection agent. $\alpha$-UMi (Shen et al., 2024a) decomposes the agent capabilities into three roles, each role is implemented by a single LLM that focuses on a specific capability and collaborates with others to accomplish the task. However, multiple models pose high requirements on computing resources, both for training and inference.

### 2.2. Parameter Efficient Fine-tuning

Parameter-efficient fine-tuning (PEFT) has emerged as a crucial technique for adapting large pre-trained models to downstream tasks while minimizing computational costs and preserving model performance. LoRA (Hu et al., 2021) freezes the pretrained model weights and injects trainable rank decomposition matrices into each layer of the Transformer architecture, greatly reducing the number of trainable parameters. LoRA+ (Hayou et al., 2024) corrects the suboptimality of LoRA by setting different learning rates for the LoRA adapter matrices with a well-chosen fixed ratio. DoRA (Liu et al., 2024b) decomposes the pre-trained weight into two components *i.e.*, magnitude and direction,

for fine-tuning, specifically employing LoRA for directional updates. QLoRA (Dettmers et al., 2024) further reduces memory use without sacrificing performance with 4-bit NormalFloat, double quantization and paged optimizers.

To further improve the accuracy, researchers recently pay more attention to using multiple LoRA combinations. LoRAHub (Huang et al., 2023) and MoA (Feng et al., 2024) pioneer the approach of training several LoRA weights on downstream tasks and then integrating the LoRA modules into a shared LLM using a routing mechanism. MoLE (Wu et al., 2024) treats each layer of trained LoRAs as a distinct expert and implements hierarchical weight control by integrating a learnable gating function to learn optimal composition weights. OCTAVIUS (Chen et al., 2023) design a multimodal LoRA-MoE decoder for modality-specific learning. MoLA (Gao et al., 2024) introduces a method of layer-wise expert allocation, where each model layer has the flexibility to employ a varying number of LoRA experts. However, these studies overlook the characteristics of agent tasks, making them challenging to apply in practice.

## 3. Method

In this section, we first introduce the decomposition of capabilities in agent tasks, then we elaborate the framework of mixture-of-roles and the objectives adopted in fine-tuning. The pipeline of preparing multi-role data is discussed at the end.

### 3.1. Capabilities Decomposition

ReAct (Yao et al., 2022) introduces a paradigm of *Reason+Action* for LLM inference, which has gradually become dominant in agent tasks (Qin et al., 2023; Gou et al., 2023; Shen et al., 2024a; Lu et al., 2024; Wang et al., 2024). Given system prompt $p$, user query $q$ and previous execution trajectory $\tau$, the LLM with weights $W$ outputs thought $T_t$ and action $A_t$ in sequence at $t$-th step:

$$T_t, A_t = W(p, q, \tau_{t-1}), \tag{1}$$

where $\tau_{t-1} = \{T_1, A_1, O_1, ..., T_{t-1}, A_{t-1}, O_{t-1}\}$ represents the execution trajectory before $t$-th step. Action consists of two parts: function name and the corresponding input parameters. $O_t$ indicates the observation of the action $A_t$ when invoked in a real deployment environment. Learning multiple capabilities simultaneously is challenging for the weights of LLM, especially for low-rank forms. Therefore, we first propose to decompose the agent capability into three roles.

**Reasoner:** The reasoner starts to understand the user's query and generate its analytical reasoning, which is then passed to the executor. Additionally, based on the observation returned after invoking tools in the real deployment

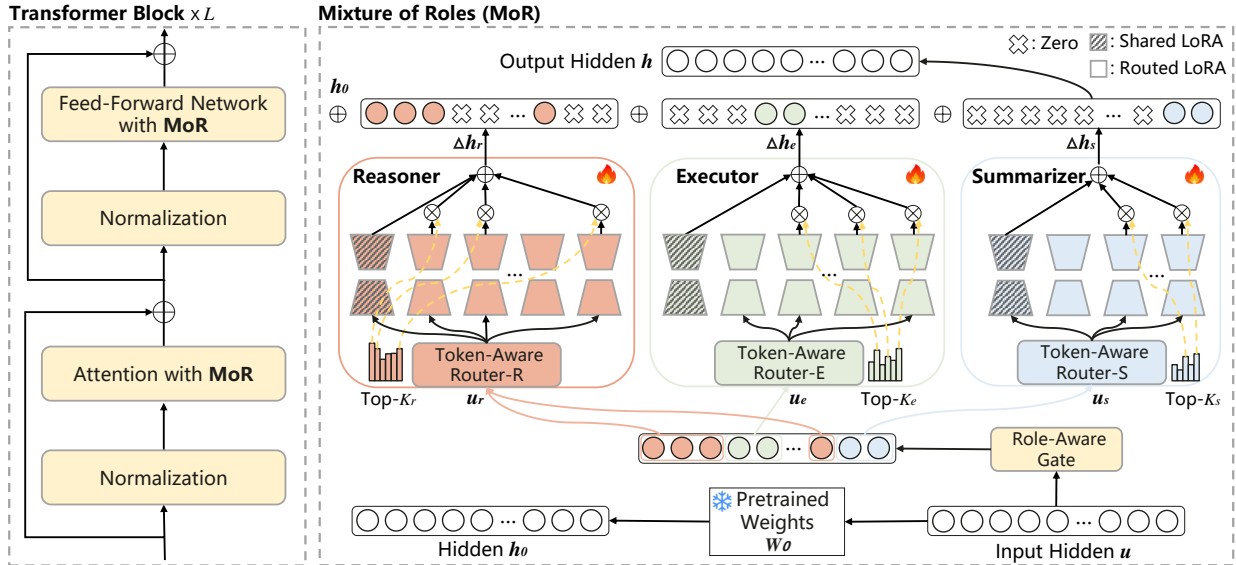

*Figure 2.* The framework of our method. The capabilities necessary for agent are decomposed into three distinct roles: reasoner, executor, and summarizer. Each role consists of different number of LoRAs according to their learning difficulty. The rule-based role-aware gate and learnable token-aware routers are introduced to more reasonably allocate LoRAs.

environment, the reasoner analyzes the execution trajectory to determine whether the user's query has been addressed. If the user query is solved, or if it cannot be solved after enough attempts, the next role is transferred to summarizer. Otherwise, it is returned to the executor for further action. Formally,

$$T_t, Role_t = \boldsymbol{W}_r(p_r, q, \tau_{t-1}), \tag{2}$$

where $Role_t$ denotes the activated role in the next step, $\boldsymbol{W}_r$ and $p_r$ denote the weights of reasoner and the system prompt of reasoner, respectively.

**Executor:** The executor is responsible for selecting the appropriate functions and parameters to invoke, based on the analytical thought of reasoner.

$$Fun_t, Param_t = \boldsymbol{W}_e(p_e, q, \tau_{t-1}, T_{t-1}), \tag{3}$$

where $Fun_t$ and $Param_t$ represent the functions and parameters in the next step, $\boldsymbol{W}_e$ and $p_e$ denote the weights and the system prompt of executor. Then, the functions are executed in the real deployment environment, such as RapidAPI Hub (Qin et al., 2023), Python IDE, *etc*.

**Summarizer:** The summarizer needs to re-organize the conversations history and convey the distilled information $Sum$ back to the user.

$$Sum = \boldsymbol{W}_s(p_s, q, \tau), \tag{4}$$

where $\boldsymbol{W}_s$ and $p_s$ denote the weights and the system prompt of summarizer, respectively.

### 3.2. Mixture-of-Roles

To characterize the decomposed capabilities, we then propose the framework of Mixture-of-Roles (MoR) as illustrated in Figure 2.

**Forward Procedure.** The MoR framework can be equipped on the linear layer of attention or feed-forward network in each transformer block. While freezing the pretrained weights in the backbone, we fine-tune only the parameters in MoR, including LoRAs and routers. Given input hidden $\boldsymbol{u} \in \mathbb{R}^{len \times d_1}$ and the frozen pretrained weights $\boldsymbol{W}_0 \in \mathbb{R}^{d_1 \times d_2}$, where $len$ is the sequence length of input, $d_1$ and $d_2$ denote the dimension of input and output, respectively. The final output hidden $\boldsymbol{h}$ can be calculated by:

$$\boldsymbol{h} = \boldsymbol{h}_0 + \Delta\boldsymbol{h}, \tag{5}$$

where $\boldsymbol{h}_0 = \boldsymbol{W}_0\boldsymbol{u}$ and $\Delta\boldsymbol{h} = \Delta\boldsymbol{h}_r + \Delta\boldsymbol{h}_e + \Delta\boldsymbol{h}_s$, *i.e.*, the sum of the outputs of reasoner, executor and summarizer. It should be noted that there is only one role that is non-zero, and the other two are all zeros. Formally, $\forall i \in \{0, ..., len - 1\}$:

$$\mathbb{1}\{\Delta\boldsymbol{h}_r^i \neq 0\} + \mathbb{1}\{\Delta\boldsymbol{h}_e^i \neq 0\} + \mathbb{1}\{\Delta\boldsymbol{h}_s^i \neq 0\} = 1. \tag{6}$$

That is, $\boldsymbol{u}[i, :]$ is processed by only one active role, which is achieved through the rule-based role-aware gate. Specifically, when a user inputs a query, the reasoner is activated first. The next role to be activated is determined based on the output of the reasoner, *i.e.*, $Role_t$ in Equation. 2. Besides, the observations $O_t$ in Equation. 1 is always input to the reasoner to decide whether to continue execution or summarize.

Without loss of generality, taking the output of reasoner as an example: $\Delta \boldsymbol{h}_r = \Delta \boldsymbol{W}_r \boldsymbol{u}_r$, where $\Delta \boldsymbol{W}_r$ denotes the LoRAs' weights of reasoner, $\boldsymbol{u}_r$ denotes the input that is allocated to reasnoer.

$$\Delta \boldsymbol{W}_r = \boldsymbol{B}_r^0 \boldsymbol{A}_r^0 + \sum_{i=1}^{E_r} \boldsymbol{B}_r^i \boldsymbol{A}_r^i \boldsymbol{R}_r(\boldsymbol{u}_r), \qquad (7)$$

Each role consists of a shared LoRA (superscript 0) and a number of $E_r$ routed LoRAs (superscript from 1 to $E_r$). $\boldsymbol{B}_r^0, \boldsymbol{B}_r^i \in \mathbb{R}^{d_1 \times d_3}$, $\boldsymbol{A}_r^0, \boldsymbol{A}_r^i \in \mathbb{R}^{d_3 \times d_2}$, and the rank $d_3 \ll \min(d_1, d_2)$. $\boldsymbol{R}_r$ denotes the Top-K token-aware router in reasoner to select the specific LoRAs for different input.

**Training Objective.** We adopt the following objective to guide the processs of supervised fine-tuning:

$$\mathcal{L}_{\text{total}} = \mathcal{L}_{\text{CE}} + \alpha_1 \mathcal{L}_{\text{aux}} + \alpha_2 \mathcal{L}_{\text{orth}}, \qquad (8)$$

where $\mathcal{L}_{\text{CE}}$ is the cross-entropy loss, which measures the difference of distribution between predication and truth. $\mathcal{L}_{\text{aux}}$ is the auxiliary balancing loss to avoid the load imbalance among LoRAs. $\mathcal{L}_{\text{orth}}$ is the orthogonal loss between LoRAs, which makes the LoRAs as independent as possible to capture features in different directions. $\alpha_1$ and $\alpha_2$ are hyper-parameters.

$$\mathcal{L}_{\text{CE}} = -\sum_{i=1}^{\mathcal{B}} \log p_w(\boldsymbol{y}^i | \boldsymbol{u}^i), \qquad (9)$$

where $\mathcal{B}$ is the batch size, $\boldsymbol{y}$ is the label for input $\boldsymbol{u}$, and $p_w$ indicates the distribution of predication with weights $w$. The auxiliary balancing loss is inherited from Switch transformers (Fedus et al., 2022), $\forall \triangledown \in \{r, e, s\}$,

$$\mathcal{L}_{\text{aux}} = E_{\triangledown} \cdot \sum_{i=1}^{E_{\triangledown}} f_{\triangledown}^i \cdot P_{\triangledown}^i, \qquad (10)$$

where $f_{\triangledown}^i$ represents the fraction of tokens dispatched to $i$-th LoRA in reasoner, executor and summarizer,

$$f_{\triangledown}^i = \frac{1}{|\mathcal{B}|} \sum_{\boldsymbol{u}_{\triangledown} \in \mathcal{B}} \mathbb{1}\{R_{\triangledown}(\boldsymbol{u}_{\triangledown})^i \neq 0\}, \qquad (11)$$

and $P_{\triangledown}^i$ denotes the average fraction of the router probability allocated for $i$-th LoRA in role $\triangledown$:

$$P_{\triangledown}^i = \frac{1}{|\mathcal{B}|} \sum_{\boldsymbol{u}_{\triangledown} \in \mathcal{B}} \text{Softmax}\left(\boldsymbol{u}_{\triangledown} \cdot \boldsymbol{W}_R^{\triangledown}\right)^i, \qquad (12)$$

where $\boldsymbol{W}_R^{\triangledown}$ denotes the weights of router in different roles. In addition, to encourage LoRAs to learn distributions in different directions and reduce redundancy, we propose the orthogonal loss.

$$L_{\text{orth}} = \sum_{i=1}^{E_{\triangledown}} \sum_{j=i+1}^{E_{\triangledown}} \left( \left\| \boldsymbol{A}_{\triangledown}^{i\,T} \boldsymbol{A}_{\triangledown}^j \right\|_F^2 + \left\| \boldsymbol{B}_{\triangledown}^{i\,T} \boldsymbol{B}_{\triangledown}^j \right\|_F^2 \right), \qquad (13)$$

---

**Algorithm 1** Fine-tuning LLM with MoR for agent tasks.

**input** A pre-trained LLM $M$, data $\mathcal{D}$ and iterations $T$ for fine-tuning, hyper-parameters $\alpha_1$ and $\alpha_2$ in loss, the modules need fine-tuning in $M$, the rank value $d_3$, the number of LoRAs $E$ in each role.

**repeat**
    **1.** Randomly select a batch of data from $\mathcal{D}$.
    **2.** Conduct the forward process for $M$ accompanied with MoR by Equation 5.
    **3.** Compute the loss function by Equation 8.
    **4.** Freeze all parameters in $M$, update the parameters of LoRAs and routers.
**until** iterations $T$.

**output** A fine-tuned LLM with MoR for agent tasks.

---

where $F$ represents the frobenius norm. Through appropriate hyper-parameters to combine different losses, better results can be achieved. The complete fine-tuning process is illustrated in Algorithm 1.

### 3.3. Data Preparation

To effectively fine-tune the framework, we develop a multi-role data generation pipeline based on publicly available datasets, incorporating role-specific content completion and reliability verification.

**Role-Specific Content Completion.** We adopt the publicly available datasets including ToolBench (Qin et al., 2023), the combination of APIGen (Liu et al., 2024d) and ToolACE (Liu et al., 2024c) and glaive-function-calling-v2 [1], MathGenie (Lu et al., 2024) to fine-tune the corresponding downstream agent tasks respectively. For ToolBench, which contains implicit multi-role content, we can directly convert it into an explicit multi-role format using rule-based methods. However, many samples in other datasets lack multi-role content, such as thought and summary. For these samples, we prompt GPT4o to complete them. The detailed prompts are presented in Figure 6 and Figure 7. In practice, we observe many execution trajectories in which the reasoning or summarizing steps are of low quality, a common issue in the outputs of LLM. To address this, we prompt DeepSeek-V3 (Liu et al., 2024a) to evaluate both the overall trajectory and the individual reasoning or summarization steps.

Further, we unify the data into JSON format to facilitate downstream fine-tuning. Specifically, the JSON begins with a list of candidate functions, if available. Each function is characterized by its name, description, parameters, and whether it is marked as required. This is followed by the system prompt and a detailed execution trajectories. The ex-

---

[1] https://huggingface.co/datasets/glaiveai/glaive-function-calling-v2

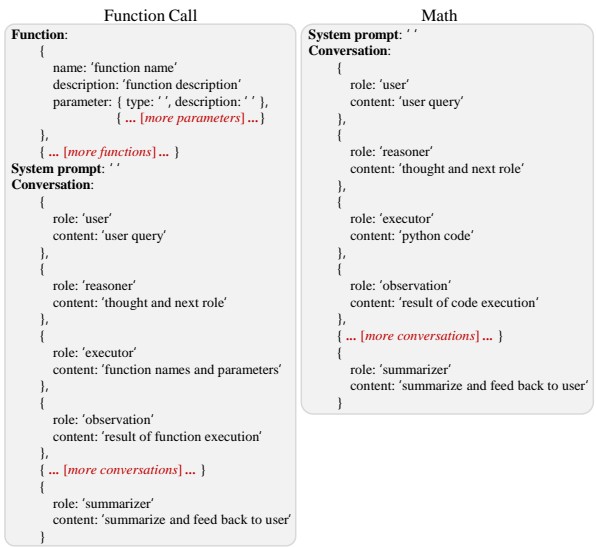

*Figure 3.* The JSON format on different scenarios in our fine-tuning datasets.

ecution trajectories starts with the user's query and proceeds with the interaction between the reasoner and the executor. If necessary, this includes execution results from the real deployment environment, *i.e.*, observations. Finally, it concludes with the feedback from summarizer. The JSON formats for different scenarios are shown in Figure 3.

**Reliability Verification.** Although the original publicly available datasets are carefully reviewed, we observe in practice that some samples contain wrong outputs from the executor. To enhance the effectiveness of fine-tuning data, we conduct a series of filtering and correction procedures. The errors can be categorized into the following types:1) Function not included in the candidate list. This issue can be identified using rule-based methods and corrected by re-prompting other LLM. 2) Incorrect numbers of parameters and type mismatches. These can be detected through execution failures and subsequently corrected manually. 3) Errors in function selection or parameter assignment. These cases are more subtle, as it does not trigger execution failures. We examine them by comparing the outputs from prompting other LLM with the execution results. Samples with inconsistencies are then manually reviewed and corrected.

## 4. Experiments

### 4.1. Models and Datasets

We conduct experiments on multiple pre-trained LLMs, including Llama3.2-1B-Instruct (Touvron et al., 2023), Phi3.5-mini-instruct (Abdin et al., 2024) and code-specific Qwen2.5-1.5B-Coder (Hui et al., 2024). For evaluations, the benchmarks including StableToolBench (Guo et al., 2024), BFCL (Yan et al., 2024), GSM8K (Cobbe et al., 2021) and MATH (Hendrycks et al., 2021) are evaluated. Specifically,

StableToolBench comprises a total of 765 questions spanning across 6 subtasks, evolving from ToolBench (Qin et al., 2023). It introduces a virtual API server and stable evaluation system. BFCL presents the first comprehensive evaluation on the LLMs' ability to call functions and tools. We report the results on Abstract Syntax Tree (AST) of live and non-live, Executable Function Evaluation (Exec) of non-live, Relevance of live, containing a total of 2797 questions. GSM8K and MATH are designed to be evaluated in the area of mathematics, which contain 1319 and 5000 questions, respectively. Different from directly generating the answer, we solve the math problems by importing python package and then executing it.

During fine-tuning, we choose the corresponding datasets for fine-tuning in their respective downstream tasks. Specifically, for StableToolBench, we sample 120k multi-role execution trajectories from the training set of toolbench. For BFCL, we sample 90k items from the combination of APIGen (Liu et al., 2024d) and ToolACE (Liu et al., 2024c) and glaive-function-calling-v2. For GSM8K and MATH, we adopt the 80k execution trajectories in MathGenie (Lu et al., 2024). All datasets are filtered and corrected by our multi-role content completion and reliability verification. The detailed prompts for system and roles in our data are illustrated in Figure 8, Figure 9 and Figure 10.

### 4.2. Implementation Details

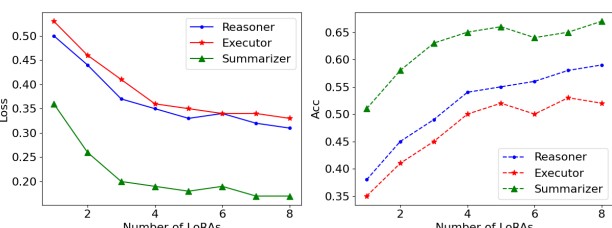

*Figure 4.* The loss and levenshtein accuracy of respective roles with different number of LoRAs.

To determine the optimal number of LoRAs per role, aiming to balance performance and the number of trainable parameters, we conduct toy experiments using the specific capability dataset of each role. The dataset is constructed by first combining all multi-role data and then splitting it based on roles. For each role, 80k samples are randomly selected as the training set, while 5k samples are sampled as the validation set. We set the dimension $d_3$ in Equation 7 to 16 and vary the number of LoRAs for each role from 1 to 8. The modules including query, key, value, out, gate, up, and down are selected as target modules for fine-tuning with MoR. Additionally, routers are removed for simplicity. The loss and levenshtein accuracy after 2 epoch fine-tuning based on Llama3.2-1B-Instruct are shown in Figure 4. We can observe that the trend of loss and accuracy changes

*Table 1.* The pass rate and win rate of different LLMs on StableToolBench.

| Model | Setting | I1-Inst | | I1-Tool | | I1-Cat | | I2-Inst | | I2-Cat | | I3-Inst | | AVG | |
|---|---|---|---|---|---|---|---|---|---|---|---|---|---|---|---|
| | | Pass | Win | Pass | Win | Pass | Win | Pass | Win | Pass | Win | Pass | Win | Pass | Win |
| GPT4 | CoT | 52.8 | 48.4 | 51.9 | 40.3 | 56.6 | 54.2 | 52.8 | 36.6 | 51.9 | 47.6 | 52.5 | 45.2 | 53.1 | 45.4 |
| | DFS | 59.2 | 50.1 | 65.7 | 44.2 | 61.7 | 57.9 | 55.2 | 41.2 | 55.6 | 53.7 | 66.1 | 45.4 | 60.6 | 48.8 |
| ToolLLaMA-v2-7B | CoT | 51.8 | 39.2 | 46.4 | 33.1 | 53.1 | 39.9 | 48.9 | 31.3 | 51.6 | 40.7 | 37.2 | 42.2 | 48.2 | 37.7 |
| | DFS | 61.0 | 43.6 | 45.6 | 35.8 | 58.8 | 41.3 | 53.5 | 34.2 | 60.3 | 43.8 | 48.1 | 45.1 | 54.6 | 40.6 |
| *Base Model: Llama3.2-1B-Instruct (1.24B)* | | | | | | | | | | | | | | | |
| Base | CoT | 14.4 | 23.3 | 10.1 | 15.8 | 12.2 | 15.0 | 3.9 | 11.3 | 5.9 | 11.3 | 8.2 | 13.1 | 9.1 | 15.0 |
| | DFS | 11.7 | 20.8 | 11.3 | 17.1 | 14.6 | 18.3 | 8.5 | 9.4 | 2.3 | 12.9 | 11.5 | 11.5 | 10.0 | 15.0 |
| **MoRAgent-Llama** | CoT | 54.9 | 40.5 | 58.2 | 38.0 | 53.1 | 39.9 | 38.1 | 34.0 | 42.6 | 41.1 | 47.9 | 29.5 | 49.1 (**+40.0**) | 37.2 (**+22.2**) |
| (1.24B+0.16B) | DFS | 54.6 | 44.8 | 45.5 | 32.9 | 53.2 | 40.5 | 46.8 | 36.8 | 68.2 | 46.8 | 58.8 | 39.4 | 54.5 (**+44.5**) | 40.2 (**+25.2**) |
| *Base Model: Phi3.5-mini-Instruct (3.82B)* | | | | | | | | | | | | | | | |
| Base | CoT | 29.7 | 30.7 | 37.9 | 30.4 | 36 | 34.6 | 20.9 | 17.0 | 26.7 | 25.0 | 17.8 | 15.8 | 28.2 | 25.6 |
| | DFS | 46.4 | 35.0 | 50.6 | 30.4 | 51.1 | 49.0 | 39.9 | 30.2 | 41.5 | 27.4 | 37.6 | 32.8 | 44.5 | 34.1 |
| **MoRAgent-Phi** | CoT | 51.8 | 47.2 | 54.4 | 38.6 | 55.4 | 51.0 | 48.1 | 33.0 | 55.0 | 45.2 | 62.3 | 42.6 | 54.5 (**+26.3**) | 42.9 (**+17.3**) |
| (3.82B+0.36B) | DFS | 55.9 | 48.5 | 56.6 | 41.1 | 60.9 | 47.1 | 55.4 | 44.3 | 59.7 | 48.4 | 63.6 | 50.8 | 58.7 (**+14.2**) | 46.7 (**+12.6**) |

slows down significantly when the number of LoRAs for both the reasoner and the executor is changed from 4 to 5. While for summarizer, it occurs at 3 to 4. Accordingly, in subsequent experiments, we set the number of activated LoRAs to 4 and the total number of LoRAs to 5 for reasoner and executor, and set the number of activated LoRAs to 3 and the total number of LoRAs to 4 for summarizer. This is also in line with our perception that the summarizer's primary task is to convey the distilled information from the historical conversation to the user, making its learning process relatively less challenging. Also, we set the learning rate to 5e-5, with 4 epochs of fine-tuning by MoR, and $\alpha_1$ and $\alpha_2$ in Equation 8 set to 1e-3 and 1e-4, respectively.

### 4.3. Experimental Results

**Results on StableToolBench.** We first evaluate the proposed method with various LLMs on StableToolBench (Guo et al., 2024). Specifically, we report the solvable pass rate and solvable win rate based on the solvable tasks with two settings including Chain-of-Thought (CoT) and Depth-First-Search (DFS). For win rate, we run all models once against GPT3.5-1106+CoT and evaluate them three times. We respectively adopt two LLMs including Llama3.2-1B-Instruct and Phi3.5-mini-Instruct as our base models. As reported in Table 1, for Llama3.2-1B-Instruct, with only introduced 0.16B trainable parameters, we improve the pass rate and win rate by 44.5% and 25.2% on DFS setting respectively. Also, for Phi3.5-mini-Instruct, with only introduced 0.36B trainable parameters, we improve the pass rate and win rate by 26.3% and 17.3% on CoT setting respectively. With fewer parameters, our MoRAgent-Phi even surpasses the agent-specific model ToolLLaMA-v2, which validates the effectiveness of our method.

**Results on BFCL leaderboard.** We then evaluate our method on BFCL leaderboard. Specifically, We report the results on Abstract Syntax Tree (AST) of live and non-live, Executable Function Evaluation (Exec) of non-live, Rel-

*Table 2.* The performance of different LLMs on BFCL.

| Model | Non-live | | Live | | AVG |
|---|---|---|---|---|---|
| | AST | Exec | AST | Relevance | |
| Qwen2.5-72B-it | 90.8 | 92.7 | 75.3 | 100.0 | 89.7 |
| GPT4 | 88.1 | 89.4 | 79.8 | 83.3 | 85.2 |
| ToolACE-8B | 87.5 | 89.2 | 78.5 | 83.3 | 84.6 |
| MiniCPM3-4B | 80.8 | 87.6 | 70.0 | 72.2 | 77.7 |
| Llama3.2-3B-it | 80.6 | 83.7 | 55.8 | 88.9 | 77.3 |
| *Base Model: Llama3.2-1B-Instruct (1.24B)* | | | | | |
| Base | 21.9 | 19.2 | 29.8 | 38.9 | 27.5 |
| **MoRAgent-Llama** (1.24B+0.16B) | 75.2 | 80.0 | 60.7 | 94.4 | 77.6 (**+50.1**) |
| *Base Model: Phi3.5-mini-Instruct (3.82B)* | | | | | |
| Base | 69.0 | 57.3 | 58.2 | 72.2 | 64.2 |
| **MoRAgent-Phi** (3.82B+0.36B) | 83.0 | 78.2 | 72.7 | 94.4 | 82.1 (**+17.9**) |

evance of live. We also adopt Llama3.2-1B-Instruct and Phi3.5-mini-Instruct as our base models. As reported in Table 2, with only introduced 0.16B and 0.36B trainable parameters, we improve the average accuracy of Llama3.2-1B-Instruct and Phi3.5-mini-Instruct by 50.1% and 17.9%, respectively.

*Table 3.* The performance of different LLMs on GSM8K and MATH.

| Model | GSM8K | MATH |
|---|---|---|
| Llama-3.1-8B-it | 54.8 | 21.4 |
| Llama-3.2-1B-it | 52.6 | 29.2 |
| DeepSeekMath-7B | 64.2 | 36.2 |
| *Base Model: Qwen2.5-1.5B-Coder (1.54B)* | | |
| Base (w/ packages) | 54.7 | 23.5 |
| Base (w/o packages) | 41.7 | 33.5 |
| **MoRAgent-Qwen** (1.54B+0.27B) | 68.5 (**+13.8**) | 45.5 (**+12.0**) |

**Results on GSM8K and MATH.** We also extend our approach to the mathematical scenarios. Different from directly generating the answer, we solve the math problems by importing python package and then executing it. We use Qwen2.5-1.5B-Coder as the base model and the results are reported in Table 3. Specifically, we adopt the framework of ToRA (Gou et al., 2023) to infer and evaluate. For

*Table 4.* Analysis on each loss functions.

| loss function | | | AVG |
|---|---|---|---|
| cross-entropy | balance | orthogonal | Acc |
| ✓ | | | 72.3 |
| ✓ | ✓ | | 75.1 |
| ✓ | | ✓ | 74.4 |
| ✓ | ✓ | ✓ | 77.6 |

*Table 5.* Analysis on the number of LoRAs.

| Total LoRAs | | | Trainable | AVG |
|---|---|---|---|---|
| reasoner | executor | summarizer | Params | Acc |
| 4 | 4 | 3 | 0.13B | 73.2 |
| 5 | 5 | 4 | 0.16B | 77.6 |
| 6 | 6 | 5 | 0.19B | 79.1 |
| 7 | 7 | 6 | 0.23B | 80.0 |

*Table 6.* The accuracy of different fine-tuning methods.

| Method | Trainable Params | Non-live | | Live | | AVG Acc |
|---|---|---|---|---|---|---|
| | | AST | Exec | AST | Rele | |
| Base | - | 21.9 | 19.2 | 29.8 | 38.9 | 27.5 |
| LoRA | 0.16B | 59.7 | 64.2 | 56.3 | 81.8 | 65.5 (+38.0) |
| DoRA | 0.16B | 61.2 | 65.7 | 58.4 | 82.0 | 66.8 (+39.3) |
| SFT | 1.24B | 72.3 | 77.6 | 61.5 | 92.6 | 76.0 (+48.5) |
| Ours | 0.16B | 75.2 | 80.0 | 60.7 | 94.4 | 77.6 (+50.1) |

the Qwen2.5-1.5B-Coder base model, we evaluate both the methods of with and without python packages. From the results, with only introduced 0.27B trainable parameters, compared with the best accuracy of the two baseline methods, we improve by 13.8% and 12.0% on GSM8K and MATH respectively.

## 4.4. Ablation Studies

We conduct a series of ablation studies in this sub-section, the Llama3.2-1B-Instruct is adopted as the base model, and the results on BFCL leaderboard are repoted.

**Loss Functions.** We first analyze each loss functions in Equation 8, the results are reported in Table 4. For the loss of balance and orthogonal, the introduction of each loss can bring about an improvement in average accuracy. The combination of them can bring a 5.3% improvement. We visualize the similarity of routed LoRAs without and with orthogonal loss in Figure 5. The multiplied weights $B$ and $A$ of the query module at layer 7 are used for visualization. With the introduction of orthogonal loss, we encourage different LoRAs to learn features in different directions, thereby reducing parameter redundancy and improving the performance of downstream tasks.

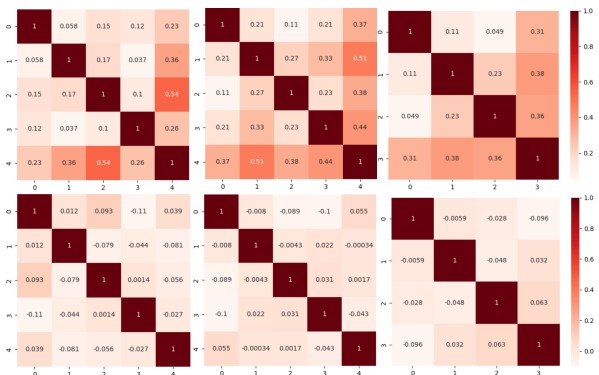

*Figure 5.* Visualization of the similarity of routed LoRAs without (top) and with (bottom) orthogonal loss. From left to right are reasoner, executor and summarizer.

**Number of LoRAs.** We also explore the influence of different number of LoRAs for various roles. In practice, we leave only one LoRA inactive by default. From Table 5,

increasing the number of LoRAs will generally increase the performance on agent task, but will undoubtedly also increase the trainable parameters. For example, when the trainable parameters decreases from 0.16B to 0.13B, the performance decreases from 77.6% to 73.2%. When the trainable parameters increases from 0.16B to 0.19B, the performance only increases by 1.5%, and further increasing the parameters does not bring significant improvement in performance. In order to achieve a balance between trainable parameters and performance, we set the number of LoRAs to 5, 5, and 4 for reasoner, executor, and summarizers in our experiments, respectively.

**Comparisons with other methods.** We compare the proposed method with other methods including parameter-efficient and full-parameter fine-tuning with the same multi-roles dataset in Table 6. From the results, SFT exhibits superior accuracy compared to PEFT methods (LoRA and DoRA), which can be attributed to its more trainable parameters, achieving an average accuracy 10.5% higher than LoRA. Notably, DoRA (Liu et al., 2024b) introduces an advanced scheme by decomposing pretrained weight matrices into magnitude vectors (m) and directional matrices (V), where LoRA is applied specifically to V while m is trained separately. This architectural innovation allows DoRA to surpass LoRA slightly in accuracy. Crucially, our method achieves statistically significant performance improvements through two key innovations: 1) a more rational capacity decomposition strategy, and 2) a novel Mixture-of-Roles framework enabling dynamic interaction between decomposed modules. These enhancements collectively contribute to our method's marked accuracy superiority over SFT and PEFT methods.

**Number of fine-tuning samples.** In order to analyze the impact of various numbers of fine-tuning samples on accuracy, we conduct corresponding experiments here, and the results are shown in Table 7. Even with only 1k training

*Table 7.* The accuracy of different number of fine-tuning samples.

| #Samples | Non-live | | Live | | AVG |
|---|---|---|---|---|---|
| | AST | Exec | AST | Rele | Acc |
| 0 | 21.9 | 19.2 | 29.8 | 38.9 | 27.5 |
| 1K | 49.5 | 44.6 | 46.3 | 79.1 | 54.9 (+27.4) |
| 5K | 55.3 | 51.8 | 50.6 | 82.1 | 60.0 (+32.5) |
| 10K | 58.8 | 57.4 | 52.7 | 85.9 | 63.7 (+36.2) |
| 50K | 70.4 | 74.9 | 56.5 | 91.7 | 73.4 (+45.9) |
| 90K | 75.2 | 80.0 | 60.7 | 94.4 | 77.6 (+50.1) |

samples, we still achieves a 27.4% improvement in average accuracy, demonstrating its well generalization capability. As the fine-tuning data volume increases, the accuracy further improves accordingly.

## 5. Conclusion

To improve the efficiency of applying PEFT to agent, in this paper, we introduce three strategies: 1) Following the increasingly dominant *Reason+Action* paradigm, we first decompose the capabilities necessary for the agent tasks into three distinct roles: reasoner, executor, and summarizer. 2) We then propose the Mixture-of-Roles (MoR) framework, which comprises three specialized LoRA groups, each designated to fulfill a distinct role. By focusing on their respective specialized capabilities and engaging in collaborative interactions, these LoRAs collectively accomplish the overall agent task. Additionally, we introduce both a rule-based role-aware gate and learnable token-aware routers to more reasonably allocate LoRAs to the input features. During the training process, auxiliary balance loss and orthogonal loss between LoRAs are further introduced for better optimization. 3) We also develop a multi-role data generation pipeline based on publicly available datasets to effectively fine-tune the framework. We conduct extensive experiments and thorough ablation studies on various LLMs and agent benchmarks, demonstrating the effectiveness of our method.

## Impact Statement

This paper presents work whose goal is to advance the field of Machine Learning. There are many potential societal consequences of our work, none which we feel must be specifically highlighted here.

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

# A. Detailed Prompts for Completing the Content of Roles.

**Reasoner**: Your role is a reasoner with reasoning capabilities; you have the ability to analyze the current task resolution status based on user queries and conversation history, and decide which role to call next. Please note the following:

1. You do not need to directly generate task answers, you only need to analyze and think about the solution approach that can complete the task based on the current status.
2. If you believe that the executor has completed the task in the historical conversation, designate the next role as summarizer to provide a summary, and let him output the results of the executor.
3. If you believe the executor has not yet completed the task, analyze and output the correct solution approach, and designate the next role as executor to carry out a detailed resolution.

Please output in the following format: The solution approach after thoughtful analysis. Next: Choose the next role to call, either executor or summarizer. Please strictly adhere to the above format for output, and end with either "Next: executor" or "Next: summarizer".

**Executor**: Your role is an executor, and you have the capability to provide specific answers to tasks based on user queries and the analysis provided by the reasoner. If the task involves tool invocation, directly output the correct tool and its parameters in the required format. Please note the following:

1. If you feel that based on the current information, you are not yet able to output the correct answer, designate a role as reasoner to continue further reasoning and analysis.
2. Even though you are certain that you have completed the user's task, please designate the next role as reasoner and let him determine further.

Please output in the following format: The function or tool called and its parameters (if the user's query is a tool invocation task). Next: Choose the next role to call, either reasoner or summarizer. Please strictly adhere to the above format for output, and only end with "Next: reasoner".

**Summarizer**: Your role is a summarizer, and you have the ability to produce summaries based on user queries and the dialogue history. Please strictly adhere to the following rules:

1. If the task involves tool invocation and there are no actual tool invocation return observation results in the historical conversation, you can examine the correctness of the tools and parameters output by the executor, as well as the output format.
2. If you believe there are errors in the output, correct the output according to the required format.
3. If you believe the output is entirely correct, then directly copy and output the correct answer generated by the executor.
4. If the task involves tool invocation and there are actual tool invocation return observation results in the historical conversation, you can provide a concise summary of the tool invocation feedback results in natural language.
5. If the task does not involve tool invocation, you can summarize the information you have received to answer the user's queries.

Please note the format requirements for the output. As you are the final step in the conversation, there is no need to specify the next role to be called.

*Figure 6.* Detailed prompts for completing the content of roles in function-calling scenarios.

# B. Detailed Prompts for System and Roles.

# C. Visualization of Execution Trajectory.

**Reasoner**: You're a professional math problem thinking assistant. You can deduce a thought process based on a given problem and existing code solutions. Please give the thought process based on the above information. Please note that you need to follow the following rules:

1. Read and understand the math problems and code solutions above, and give answers.
2. Your thinking process needs to give a step-by-step solution to the problem, and you need to provide formulas and parameters for the code solution, but you don't need to calculate the results.
3. The first step should be to extract the variables and corresponding values from the problem, and use First, Next, and Finally as transition words between steps.
4. It's best to conclude the thought process by pointing out which parameters lead to the final result we need.
5. Formulas in the thinking process are best highlighted in parentheses.
6. The thought process you give needs to be as concise as possible. Don't use mathematical symbols elsewhere to avoid miscalculations.

*Figure 7.* Detailed prompts for completing the content of roles in mathmatical scenarios.

**System**: You have access to the following APIs within XML tags:<tools>[doc]</tools>

**Reasoner**: Your role is a reasoner with reasoning capabilities, and you have the ability to analyze the current task status based on user queries and conversation history, and decide which role to call next. Please note the following:

1. You do not need to directly generate task answers, you only need to analyze and think about the solution approach that can complete the task based on the current status.
2. If you believe that the Executor has completed the task in the historical conversation, designate the next role as summarizer to provide a summary.
3. If you believe the executor has not yet completed the task, analyze and output the correct solution approach, and designate the next role as executor to carry out a detailed solution.

Please output in the following format: The solution approach after thoughtful analysis. Next: Choose the next role to call, either executor or summarizer.

**Executor**: Your role is an executor, and you have the capability to provide specific api calling to tasks based on user queries and the analysis provided by the reasoner. Please directly output the correct api tool and its parameters with the following format: [unused11]Action: <function-name> Arguments: <args-dict>[unused12] Please strictly adhere to the above format for output.

**Summarizer**: Your role is a summarizer, and you have the ability to produce summaries based on user queries and the dialogue history. Please strictly adhere to the following rules:

1. If the task involves tool invocation and there are actual tool invocation return observation results in the historical conversation, you can provide a concise summary of the tool invocation feedback results in natural language.
2. If the task does not involve tool invocation, you can summarize the information you have received to answer the user's queries.

*Figure 8.* Detailed prompts of the datasets for fine-tuning in BFCL.

**System**: You have access to the following APIs within XML tags:<tools>[doc]</tools>

**Reasoner**: Your role is a reasoner with reasoning capabilities; you have the ability to analyze the current task status based on user queries and conversation history, and decide which role to call next. Please note the following:
1. You do not need to directly generate task answers; you only need to analyze and think about the solution approach that can complete the task based on the current status.
2. If you believe that the Executor has completed the task in the historical conversation, designate the next role as summarizer to provide a summary.
3. If you believe the executor has not yet completed the task, analyze and output the correct solution approach, and designate the next role as executor to carry out a detailed solution.

Please output in the following format: The solution approach after thoughtful analysis. Next: Choose the next role to call, either executor or summarizer. Please strictly adhere to the above format for output, and end with either ”Next: executor.” or ”Next: summarizer.

**Executor**: Your role is an executor, and you have the capability to provide specific api calling to tasks based on user queries and the analysis provided by the reasoner. Please directly output the correct api tool and its parameters with the following format: [unused11]Action: <function-name> Action-Input: <args-dict>[unused12] Please strictly adhere to the above format for output.

**Summarizer**: Your role is a summarizer, and you have the ability to produce summaries based on user queries and the dialogue history. Please strictly adhere to the following rules:
1. If the task involves tool invocation and there are actual tool invocation return observation results in the historical conversation, you can provide a concise summary of the tool invocation feedback results in natural language.
2. If the task does not involve tool invocation, you can summarize the information you have received to answer the user's queries.

*Figure 9.* Detailed prompts of the datasets for fine-tuning in ToolBench.

**System**: You have the capability to address the following user's query.

**Reasoner**: Your role is a reasoner with reasoning capabilities; you have the ability to analyze the current task status based on user queries and conversation history, and decide which role to call next. Please note the following:
1. You do not need to directly generate task answers; you only need to analyze and think about the solution approach that can complete the task based on the current status.
2. If you believe that the executor has completed the task in the historical conversation, designate the next role as summarizer to provide a summary.
3. If you believe the executor has not yet completed the task, analyze and output the correct solution approach, and designate the next role as executor to carry out a detailed solution.

Please output in the following format: The solution approach after thoughtful analysis. Next: Choose the next role to call, either executor or summarizer. Please strictly adhere to the above format for output, and end with either ”Next: executor.” or ”Next: summarizer.”

**Executor**: Your role is an executor. You have the capability to generate specific Python code based on the conversation history and the output of the executot to address the user's query. Please directly generate a segment of Python code.

**Summarizer**: Your role is a summarizer, and you have the ability to produce summaries based on user queries and the dialogue history.

*Figure 10.* Detailed prompts of the datasets for fine-tuning in Math.

**User Query**: I'm planning a small outdoor event in Ottawa, and I need to make sure the weather is going to cooperate. Could you fetch the current weather for me at latitude 45.4215 and longitude -75.6972 using the Open-Meteo API? Also, I'm running a small game at the event, and I'm curious about the chances of winning. If I have 10 attempts at this game and the chance of winning each time is 50%, how likely is it that I'll win 5 times?

**ToolList**: get-weather-data, calc-binomial-probability

**Reasoner**: I need to use the tool get-weather-data and calc-inomial-probability to solve this problem. This involves providing the following parameters:first param is {coordinates': [45.4215, -75.6972]}, second param is {'n': 10, 'k': 5, 'p': 0.5}

**Executor**: Action: get-weather-data Arguments: {'coordinates': [45.4215, -75.6972]}, Action: calc-binomial-probability Arguments: {'n': 10, 'k': 5, 'p': 0.5}

**Summarizer**: According to the historical record, I've provided you with the api for finding the weather and calculating the odds of winning games.

*Figure 11.* Visualization of execution trajectory on BFCL.

**User Query**: If $2x = 5y$ and $7y = 10z$, what is the value of $\frac{z}{x}$ expressed as a common fraction?

**Reasoner**: Alright, let's solve this step by step.Given:

$$2x = 5y$$

$$7y = 10z$$

We want to find the value of $\frac{z}{x}$. Step 1: Solve for $y$ from the first equation.From $2x = 5y$, we get:

$$y = \frac{2x}{5}$$

Step 2: Substitute the expression for $y$ into the second equation to solve for $z$. Substituting $y = \frac{2x}{5}$ into $7y = 10z$:

$$7\left(\frac{2x}{5}\right) = 10z$$

Solving for $z$:

$$z = \frac{7 \times \frac{2x}{5}}{10}$$

Step 3: Find the ratio $\frac{z}{x}$. Now, we'll calculate $\frac{z}{x}$. Next: Executor

**Executor**:

```python
from sympy import symbols, Eq, solve

# Define the variables
x, y, z = symbols('x y z')

# Given equations
eq1 = Eq(2*x, 5*y)
eq2 = Eq(7*y, 10*z)

# Solve eq1 for y
y_expr = solve(eq1, y)[0]

# Substitute y in eq2
eq2_sub = eq2.subs(y, y_expr)

# Solve for z
z_value = solve(eq2_sub, z)[0]

# Calculate z/x
ratio_z_to_x = z_value / x
print(ratio_z_to_x)
```

**Summarizer**: By solving the given equations step-by-step, we determined that the value of $\frac{z}{x}$ is $\boxed{\dfrac{7}{25}}$. This result was obtained through substitution and simplification, confirming our calculations and validating the relationship between $x$, $y$, and $z$.

*Figure 12.* Visualization of execution trajectory on Math.

