# OpenReview forum: "MoRAgent: Parameter Efficient Agent Tuning with Mixture-of-Roles"
_ICML.cc/2025/Conference — ICML 2025 poster_

### Official Review · Reviewer_Mktg · 2025-03-12

**Overall Recommendation:** 5

**Summary:**

This work presents a novel framework to fine-tune LLMs to solve agent-specific task in a parameter-efficient manner. Specifically, the capabilities of agent are firstly decomposed into three roles. A relative fine-tuning framework MoR and a multi-role data generation pipeline are subsequently proposed to ensure that LLMs can correctly learn the capabilities of different roles.

**Claims And Evidence:**

Yes, the author conducts several experiments with different size of models on different agent-specific tasks, and the results are convincing.

**Essential References Not Discussed:**

No, important related works are already cited and discussed in this work.

**Experimental Designs Or Analyses:**

Yes. The three main tables have shown to be validate, the ablation studies also prove the correctness of the hyper-parameters and the design of the loss function.

**Methods And Evaluation Criteria:**

Yes. The author selects both agent-specific and mathematical benchmarks (which could also be solved in an agentic manner) to evaluate the proposed method.

**Other Comments Or Suggestions:**

Typos:
Caption of Figure 8, ‘mathmatical’ -> ‘mathematical’

Suggestions:
The code of the architecture and training process are suggested to be released.

**Other Strengths And Weaknesses:**

Strengths:
1)	Other than directly training multi-models to learn different roles capabilities, the author introduces a novel architecture: Mixture-of-Roles, which comprises three specialized LoRA groups.
2)	The author has conducted comprehensive experiments on different benchmarks with different base models, the results show a significant improvement compared with the base models.
3)	The contributions of different parts in the loss function are studied through extensive experiments in the part of Ablation Studies.
Weakness:
1)	To prove the effectiveness of the multi-role strategy, experiments with a single role is supposed to be conducted.
2)	When adapt to each new task, the work of data preparation and unification is essential, but seems time-consuming and labor-intensive.

**Questions For Authors:**

1. Have you conducted additional experiments on larger models like 7B or 14B?
2. In table 3, why qwen performs better after training when the base results are worse than llama on MATH benchmarks.
3. Have the authors tried other role-decomposition strategies? For example, does using more or fewer roles have a greater impact on the results?

**Relation To Broader Scientific Literature:**

The related scientific literature of the main contributions of the paper are already cited and studied. For example: the idea of role decomposition may relate to alpha-UMi[1], and the novel architecture may evolve from the work of OCTAVIUS[2] and MoLA[3].
[1] Small llms are weak tool learners: A multi-llm agent.
[2] Octavius: Mitigating task interference in mllms via moe
[3] Higher layers need more lora experts.

**Theoretical Claims:**

This work does not involve theoretical claims and proofs.

---

> ### Author Rebuttal · Authors · 2025-04-01
>
> Dear Reviewer Mktg:
>
> Grateful for your support and helpful review. All concerns and questions are meticulously responded.
>
> **1. Experiments with a single role is supposed to be conducted.**
>
> Thanks for the helpful suggestion. We supplement the experiment of a single lora without roles on BFCL leaderboard with Llama3.2-1B-Instruct.
>
> |Method|Trainable Params|AST (Non-live) | Exec (Non-live) | AST (Live) | Relevance (Live) | AVG|
> | :-:| :-: | :-: | :-: | :-: |:-: |:-: |
> |Base|-|21.9 | 19.2 | 29.8 | 38.9 |27.5|
> |LoRA | 0.16B|60.5 |68.2 | 59.4|83.7|68.0 (+40.5)|
> |MoR|0.16B| 75.2 |80.0 | 60.7|94.4|77.6 (+50.1)|
>
> With the same number of trainable parameters, due to the lack of decomposition capability, the accuracy improvement achieved by LoRA is much lower than that of MoR.
>
> **2. The work of data preparation and unification is essential, but seems time-consuming and labor-intensive.**
>
> Thanks for the kind concern. Actually, a large proportion of the raw training data lacks multi-roles content, such as the thought of reasoner. Therefore, it cannot be directly used for MoR training. The data processing pipeline proposed in the paper addresses this by generating high-quality multi-role data based on the raw dataset. However, as the reasoner paradigm like DeepSeek-R1 and OpenAI o3, etc., has become mainstream recently, there will be more agent related data with thought.
>
> **3. Additional experiments on larger models.**
>
> Thanks for the helpful suggestion. Due to the limited time of rebuttal, we only supplement the experiment of Llama3.1-8B-Instruct on BFCL.
>
> |Method|Trainable Params|AST (Non-live) | Exec (Non-live) | AST (Live) | Relevance (Live) | AVG|
> | :-:| :-: | :-: | :-: | :-: |:-: |:-: |
> |Base|-| 84.2| 86.3 | 61.0 | 77.8 |77.3|
> |MoR|0.59B| 88.6 |89.2 | 80.5|95.1|88.4 (+11.1)|
>
> The experimental settings are consistent with section4.2 in our paper. From the results, with the introduction of 0.59B trainable parameters, we improve the average accuracy by 11.1%.
>
> **4. In table 3, why qwen performs better after training when the base results are worse than llama on MATH benchmarks.**
>
> The possible reason is that llama is a general model and qwen-coder is a code-specific model. Since we solve math problems by importing packages here, it is more in line with the style of qwen-coder, so the accuracy of qwen-coder can be improved higher in the post-training stage.
>
> **5. Using more or fewer roles have a greater impact on the results?**
>
> Thanks for the constructive suggestion, we supplement the experiments of more or fewer roles. The results are as follows.
>
> |Archs| Trainable Params|AST (Non-live) | Exec (Non-live) | AST (Live) | Relevance (Live) | AVG|
> | :-: | :-: | :-: | :-: | :-: |:-: |:-: |
> |Base| -|21.9 | 19.2 | 29.8 | 38.9 |27.5|
> |2-roles| 0.11B | 60.6 |66.3 | 48.3|79.5|63.7 (+36.2)|
> |3-roles|0.16B| 75.2 |80.0 | 60.7|94.4|77.6 (+50.1)|
> |4-roles|0.21B| 70.8 |74.2 | 57.6|90.3|73.2 (+45.7)|
>
> Our method integrates three core roles: Reasoner, Executor, and Summarizer. Through removing the Summarizer and introducing a Planner, we extend the roles to 2-roles and 4-roles. Experimental results demonstrate accuracy improvements across all variants compared to the baseline model. Notably, the 3-roles architecture achieves peak performance (accuracy improvement of 50.1% over baseline), whereas the 4-roles, despite containing more trainable parameters, shows diminishing returns (45.7% accuracy gain). This empirical evidence highlights that ​rational role definition outweigh mere parameter quantity expansion in achieving optimal model performance.
>
> **6. Typos.**
>
> Sincerely thanks for your detailed reviews. We will go through the entire paper again and fix potential typos in the revised version.
>
> **7. The code of the architecture and training process are suggested to be released.**
>
> Thanks, all our data and code will be open-sourced to contribute to the community.

---

### Official Review · Reviewer_rpMC · 2025-03-12

**Overall Recommendation:** 3

**Summary:**

The paper introduces a novel parameter-efficient fine-tuning method to enhance LLMs for agent tasks, such as function-calling and mathematical reasoning. The authors propose three main strategies: (1) decomposing agent capabilities into three roles—reasoner, executor, and summarizer—based on the Reason+Action paradigm; (2) developing the MoR framework, which assigns specialized LoRA groups to each role, incorporating a rule-based role-aware gate and token-aware routers to manage role interactions; and (3) creating a multi-role data generation pipeline that enhances publicly available datasets with role-specific content and reliability verification. The method is evaluated on benchmarks like StableToolBench, BFCL, GSM8k, and MATH. The paper claims that MoRAgent achieves competitive performance with fewer trainable parameters than traditional methods.

## update after rebuttal
Thank you for the response. I will keep my score.

**Claims And Evidence:**

- The paper claims that decomposing agent capabilities into three roles improves PEFT for agent tasks. The improvement from MoR is clear and comprehensive analysis is conducted to demonstrate that.
- While we do see performance improvement on downstream tasks there is a lack of a direct ablation comparing MoRAgent with and without role decomposition (e.g., a single LoRA without roles), making it hard to isolate the decomposition’s specific contribution versus the multi-LoRA setup.
- Another claim is that the MoR framework with a router works. But he rule-based role-aware gate’s implementation is vague. The paper states that “the next role to be activated is determined based on the output of the reasoner”, but lacks specifics on how this is operationalized during training, weakening the claim’s clarity.
- The effect of multi-role data generation pipeline is unclear. No direct evidence compares MoRAgent’s performance with and without this pipeline versus raw data. Also it is unclear whether the performance is coming from MoR or the data pipeline.

**Essential References Not Discussed:**

N/A

**Experimental Designs Or Analyses:**

- The experimental designs are robust, I just have two concerns detailed above: the ablation between standard LoRA and MoR and the ablations for data pipeline. Both are to isolate the contribution of the MoR framework to make a stronger argument.

**Methods And Evaluation Criteria:**

The MoR framework decomposes agent tasks into reasoner, executor, and summarizer roles, Taligning with the Reason+Action paradigm cited in prior work. The proposed benchmarks are also reasonable and cover a wide variety of downstream tasks.

**Other Comments Or Suggestions:**

N/A

**Other Strengths And Weaknesses:**

- Contribution is clear and simple: it addresses resource barriers in agent fine-tuning, relevant for deploying LLMs in resource-constrained settings.
- However, the effectiveness of the method is unclear without the ablations mentioned earlier.

**Questions For Authors:**

1. How is the rule-aware gate implemented during training? Are there labels in the training data indicating which role should be active for each token? Clarification could resolve ambiguity in Section 3.2, strengthening the method’s reproducibility.
2. How's the performance for standard LoRA? Comparing this method with standard LoRA can further consolidate the argument.
3. I got a bit confused by section 3.2 and equation (6): "It should be noted that for the token at the same location, there is only one role that is non-zero" what exactly does this mean? I thought u is the hidden state, so each element in u should just be a value of the hidden state, not a token. I couldn't connect this with figure 2. Is each value in the hidden state activated by one role, or each token?

**Relation To Broader Scientific Literature:**

Builds on full-parameter fine-tuning works like ToolLLM (Qin et al., 2023) and AgentTuning (Zeng et al., 2023), which enhance LLMs for agent tasks and Inspired by Reason+Action (Yao et al., 2022) and multi-agent systems like α-UMi (Shen et al., 2024a). It is a novel extension on top of those works by making LLMs good tool users (therefore agent) while being resource efficient.

**Theoretical Claims:**

The paper does not present theoretical claims or proofs. It is empirically driven.

---

> ### Author Rebuttal · Authors · 2025-04-01
>
> Dear Reviewer rpMC:
>
> We sincerely thanks for your support and meticulous review. The concerns and questions are answered as follows.
>
> **1. How is the rule-aware gate implemented during training? Are there labels in the training data indicating which role should be active for each token?**
>
> Yes! Your deduction is correct. Our training data includes a "role" tag to indicate which role activates the subsequent tokens. Take the Math data in Figure 3 in our paper as an example. The tokens marked as "content" are activated by the role marked as "role". It should be noted that when the role is user or observation, it is also activated by reasoner. Therefore, "system_prompt", "user_query", "thought and next role" and "code execution result" are activated by reasoner. "Python code" is activated by executor. "summarize and feed back to user" is activated by summarizer. We will further clarify this in the revised version.
>
> **2. The effect of multi-role data generation pipeline is unclear. It is unclear whether the performance is coming from MoR or the data pipeline. How's the performance for standard LoRA?**
>
> Thanks for the helpful suggestion. A large proportion of the raw training data lacks multi-role content, such as the thought of reasoner. Therefore, it cannot be directly used for MoR training. The data processing pipeline proposed in the paper addresses this by generating high-quality multi-role data based on the raw dataset. A comparative example is illustrated below.
>
> Raw data:
> ```
> {
>     "role": "user"
>     "content": "I want to watch a movie tonight. Can you suggest me some movies starring Leonardo DiCaprio?"
> },
> {
>     "role": "assistant"
>     "content": "[unused11]Action: search_movies\n Arguments: {"actor": "Leonardo DiCaprio"}[unused12]"
> },
> {
>     "role": "observation"
>     "content": "{"movies": ["Inception", "The Wolf of Wall Street", "Titanic", "The Revenant", "Shutter Island"]}"
> },
> {
>     "role": "assistant",
>     "content": " "Inception", "The Wolf of Wall Street", "Titanic", "The Revenant", "Shutter Island". "
> }
>
> ```
> Multi-role data after generation pipeline:
> ```
> {
>     "role": "user",
>     "content": "I want to watch a movie tonight. Can you suggest me some movies starring Leonardo DiCaprio?"
> },
> {
>     "role": "reasoner",
>     "content": "To suggest movies starring Leonardo DiCaprio, we can use the "search_movies" tool with the input specifying the actor's name.\n Next: executor."
> },
> {
>     "role": "executor",
>     "content": "[unused11]Action: search_movies\n Arguments: {"actor": "Leonardo DiCaprio"}[unused12]"
> },
> {
>     "role": "observation",
>     "content": "{"movies": ["Inception", "The Wolf of Wall Street", "Titanic", "The Revenant", "Shutter Island"]}"
> },
> {
>     "role": "reasoner",
>     "content": "The function has been successfully executed and five movies starring Leonardo DiCaprio are returned, which can solve the user's query.\n Next: summarizer."
> },
> {
>     "role": "summarizer",
>     "content": "Sure, here are some movies starring Leonardo DiCaprio: "Inception", "The Wolf of Wall Street", "Titanic", "The Revenant", "Shutter Island". Enjoy your movies!"
> }
> ```
> All data and code will be open-sourced to contribute to the community. We supplement the experiments of LoRA in two settings: a single LoRA without roles (standard, trained on raw data) and a single LoRA with roles (multi-roles, trained on multi-role data).
>
> |Method|Trainable Params|AST (Non-live)|Exec (Non-live)|AST (Live)|Relevance (Live)| AVG|
> |:-:|:-:|:-:|:-:|:-:|:-:|:-:|
> |Base|-|21.9|19.2|29.8|38.9|27.5|
> |LoRA (standard)|0.16B|60.5 |68.2|59.4|83.7|68.0 (+40.5)|
> |LoRA (multi-roles)|0.16B|59.7|64.2|56.3|81.8|65.5 (+38.0)|
> |MoR|0.16B|75.2|80.0|60.7|94.4|77.6 (+50.1)|
>
> Interestingly, the accuracy of LoRA (multi-roles) is lower than that of standard LoRA. The possible reason is that under limited training parameters, there is interference when a single LoRA learns the multi-roles datasets. In contrast, MoR can avoid interference by the architecture of mixture-of-roles to make full use of the limited training parameters and achieve higher accuracy.
>
> **3. Confused by equation(6).**
>
> Sorry for misleading. The size of u here can be represented by [sequence_length,hidden_dim] (for simplicity, we can omit the dimension of batch size). Without loss of generality, take the size of u in one layer equal to [4096,hidden_dim] as an example. Further example, [:1024, :] of u is processed by reasoner, [1024:1536, :] of u is processed by executor, [1536:2560, :] of u is processed by reasoner, [2560:3072, :] of u is processed by executor, [3072:3584, :] of u is processed by reasoner, and [3584:, :] of u is processed by summarizer. We will further clarify this in the revised version.

---

### Official Review · Reviewer_MDgP · 2025-03-13

**Overall Recommendation:** 3

**Summary:**

This paper proposes multiple strategies to improve the efficiency of applying PEFT to agent. First, the capabilities necessary for the agent tasks are decomposed into three distinct roles: reasoner, executor, and summarizer. The Mixture-of-Roles (MoR) framework, which comprises three specialized LoRA groups, each designated to fulfill the three roles. To more reasonably allocate LoRAs to the input features, a rule-based and role-aware gate and learnable token-aware routers are designed. During the training process, auxiliary balance loss and orthogonal loss between LoRAs are further introduced for better optimization. Last but not least, a multi-role data generation pipeline is introduced.

**Claims And Evidence:**

Yes

**Essential References Not Discussed:**

No.

**Experimental Designs Or Analyses:**

Yes, I have checkd the soundness/validity of the experiments, including StableToolBench, BFCL leaderboard, GSM8K and MATH.

**Methods And Evaluation Criteria:**

Yes, the proposed method is evaluated on StableToolBench、BFCL and Math problems.

**Other Comments Or Suggestions:**

Typos: "available datasets To effectively" in Line435.

**Other Strengths And Weaknesses:**

Strengths:
* The idea of decomposing the ability of agent into three roles is interesting.
* In my knowledge, this is the first work that apply Mixture-of-LoRAs to the agent tasks.
* The composition of rule-based gate and role-aware gate is novel.
* The experiments are conducted on StableToolBench, BFCL and Math, which are sufficient.

Weaknesses:
* Some other work like [1] also introduces the idea of decomposition of agent ability, what is the difference between you?
* The auxiliary balance loss is not novel.
* Ablation experiments on the auxiliary balance loss and orthogonal loss weight coefficients are missing.
* The preparation of experimental data seems to be a lot of work.

[1] Small llms are weak tool learners: A multi-llm agent

**Questions For Authors:**

1. Will the two gates lead to an increase in FLOPs or latency?
2. What would be the result of having three separate LLMs to represent different roles?
3. With the same parameters, what is the comparison between a large LoRA and multiple small LoRAs?

**Relation To Broader Scientific Literature:**

Three key contributions in this paper:1) Decomposing the agent ability into three roles; 2) Each abilities are fulfilled by Mixture-of LoRAs; 3) A lot of work of preparing the CoT agent training data. Any relevant previous work has been discussed in the paper.

**Theoretical Claims:**

No proofs and theoretical claims.

---

> ### Author Rebuttal · Authors · 2025-04-01
>
> Dear Reviewer MDgP:
>
> We deeply appreciate your support and insightful feedback, detailed rebuttals to all queries are provided.
>
> **1. The difference between ours and α-UMi.**
>
> α-UMi decomposes the agent ability into planner,executor and summarizer. However, each role is implemented by a separate LLM, which results in a significant increase in computing resources. In contrast, we integrate the multiple agent capabilities into a novel parameter-efficient Mixture-of-Roles framework.
>
> **2. The auxiliary balance loss is not novel.**
>
> Thanks for the kind concern. Auxiliary balance loss is widely adopted in training MoE architectures. We borrow the idea of ​​balance loss and are the first to apply it to our agent work. The experimental results on various benchmark verify the effectiveness of the loss.
>
> **3. Ablation on the auxiliary balance loss and orthogonal loss weight coefficients.**
>
> Thanks for the helpful advice. we supplement the ablation studies on BFCL leaderboard with Llama3.2-1B-Instruct.
>
> |balance loss|orthogonal loss|AST (Non-live) | Exec (Non-live) | AST (Live) | Relevance (Live) | AVG|
> | :-: | :-: | :-: | :-: |:-: |:-: |:-: |
> |1e-3|1e-4 | 75.2 |80.0 | 60.7|94.4|77.6|
> |1e-4|1e-4 | 74.6 |78.8 | 58.9|93.6|76.5|
> |1e-3|1e-3 | 75.7 |79.3 | 59.1|94.0|77.0|
> |1e-4|1e-3 | 74.1 |78.4 | 59.6|93.1|76.3|
>
> With appropiate coefficients of auxiliary balance loss and orthogonal loss, the accuracy may be further improved.
>
> **4. The preparation of experimental data seems to be a lot of work.**
>
> Actually, a large proportion of the raw training data lacks multi-role content, such as the thought of reasoner. Therefore, it cannot be directly used for MoR training. The data processing pipeline proposed in the paper addresses this by generating high-quality multi-role data based on the raw dataset. However, as the reasoner paradigm has become mainstream recently, there will be more agent related data with thought. All data and code will be open-sourced to contribute to the community.
>
>
> **5. Will the two gates lead to an increase in FLOPs or latency?**
>
> Thanks. Our gates include the rule-based role-aware gate and learnable token-aware gate. For the rule-based gate, a "role" tag is designed to indicate which role activates the subsequent tokens. Therefore, no FLOPs or latency is introduced. For each learnable token-aware gate, we implement it with only a single Linear layer, resulting in a negligible FLOPs and latency.
>
> **6. The result of having three separate LLMs to represent different roles.**
>
> This is an interesting question, and actually, that is precisely how α-UMi operates. We supplement this experiment on BFCL leaderboard with Llama3.2-1B-Instruct.
> |Method| Trainable Params|AST (Non-live) | Exec (Non-live) | AST (Live) | Relevance (Live) | AVG|
> | :-: | :-: | :-: | :-: | :-: |:-: |:-: |
> |Base| -|21.9 | 19.2 | 29.8 | 38.9 |27.5|
> |MoR|0.16B| 75.2 |80.0 | 60.7|94.4|77.6 (+50.1)|
> |3 LLMs|3.72B| 79.3|83.2 | 69.5|94.8|81.7 (+54.2)|
>
> With more trainable parameters, three separate LLMs achieve higher accuracy, but the computational resources required for training and inference are much higher than ours.
>
> **7. The comparison between a large LoRA and multiple small LoRAs?**
>
> Sorry for not fully understanding what multiple small LoRAs meant, we guess it should be our proposed MoR. We supplement the experiment of one large LoRA with Llama3.2-1B-Instruct on BFCL.
>
> |Method|Trainable Params|AST (Non-live) | Exec (Non-live) | AST (Live) | Relevance (Live) | AVG|
> | :-:| :-: | :-: | :-: | :-: |:-: |:-: |
> |Base|-|21.9 | 19.2 | 29.8 | 38.9 |27.5|
> |LoRA| 0.16B| 59.7 |64.2 | 56.3|81.8|65.5 (+38.0)|
> |MoR|0.16B| 75.2 |80.0 | 60.7|94.4|77.6 (+50.1)|
>
> The LoRA is trained on the same 90k multi-roles data. From the results, the proposed MoR can fully utilize the limited training parameters and achieve significantly better accuracy.
>
> **8. Typos.**
>
> Sincerely thanks for your detailed reviews. We will go through the entire paper again and fix potential typos in the revised version.

---

### Official Review · Reviewer_QZb4 · 2025-03-13

**Overall Recommendation:** 3

**Summary:**

This paper explores parameter-efficient fine-tuning (PEFT) methodologies for large language model (LLM)-based agent tasks, an area that remains largely unexplored. The authors propose three key strategies:

1.	Role Decomposition: Inspired by the Reason+Action paradigm, the authors decompose agent capabilities into three distinct roles—reasoner, executor, and summarizer. The reasoner interprets user queries and determines the next step based on execution trajectory. The executor identifies and invokes appropriate functions with the correct parameters. The summarizer distills and conveys information back to the user.

2.	Mixture-of-Roles (MoR) Framework: The authors introduce a framework with three specialized Low-Rank Adaptation (LoRA) modules, each dedicated to a specific role. These modules collaboratively perform the agent task while maintaining parameter efficiency.

3.	Multi-Role Data Generation Pipeline: A novel data generation pipeline is designed using publicly available datasets. It incorporates role-specific content completion and reliability verification to support fine-tuning the MoR framework.

The paper presents extensive experiments and ablation studies on various LLMs and agent benchmarks, demonstrating the effectiveness of the proposed approach.

**Claims And Evidence:**

yes

**Essential References Not Discussed:**

no

**Experimental Designs Or Analyses:**

yes

**Methods And Evaluation Criteria:**

yes

**Other Comments Or Suggestions:**

see Question

**Other Strengths And Weaknesses:**

Strengths:

1.	Novel PEFT Approach for Agents: The paper addresses a relatively unexplored area—parameter-efficient fine-tuning (PEFT) for agent tasks—by introducing a structured role-based approach.

2.	Clear Role Decomposition: The division of agent capabilities into reasoner, executor, and summarizer aligns well with the Reason+Action paradigm, making the framework interpretable and modular.

3.	Efficient Fine-Tuning via LoRA: The use of Mixture-of-Roles (MoR) with specialized LoRA modules enables efficient adaptation of LLMs without full model fine-tuning, reducing computational overhead.

Weaknesses:

1.	Limited Comparison with Other PEFT Methods: While the paper focuses on its novel approach, it lacks a direct comparison with other existing PEFT techniques that might be adapted for agent tasks.

2.	Scalability and Generalization: The approach is tailored to a specific role-based agent structure, and its effectiveness for more complex or diverse agent architectures remains unclear.

3.	Data Dependence: The proposed multi-role data generation pipeline relies on publicly available datasets, but its adaptability to real-world or unseen tasks is not extensively analyzed.

4. Limited experiments about the ablation. It is unclear how much gains contributed from the SFT on the such diverse datasets.

**Questions For Authors:**

1. What's the gain by SFT on the same datasets, w/o lora? and other PEFT methods?

2. Could you validate your approach on more agent structures? such as Reflection, AutoGen?

3. Could you validate your approach on more agent-related datasets? The training data is tool-use and the validation set is also tool-using (except for math), how about generalization ability on other agent tasks, maybe refer to Agentgym.

**Relation To Broader Scientific Literature:**

good

**Theoretical Claims:**

yes

---

> ### Author Rebuttal · Authors · 2025-04-01
>
> Dear Reviewer QZb4:
>
> Sincerely thank you for your constructive review, the concerns and questions are answered in detail.
>
> **1. What's the gain by SFT on the same datasets, w/o lora? and other PEFT methods?**
>
> Thanks for the helpful suggestion. Based on the same multi-roles dataset, we supplement the experimental results of model Llama3.2-1B-Instruct on BFCL leaderboard. The results are as follows.
>
> |Method|Trainable Params|AST (Non-live)|Exec (Non-live)|AST (Live)|Relevance (Live)|AVG|
> |:-:|:-:|:-:|:-:|:-:|:-:|:-:|
> |Base|-|21.9|19.2|29.8|38.9|27.5|
> |SFT|1.24B|72.3|77.6|61.5|92.6|76.0 (+48.5)|
> |LoRA|0.16B|59.7|64.2|56.3|81.8|65.5 (+38.0)|
> |DoRA|0.16B|61.2|65.7|58.4|82.0|66.8 (+39.3)|
> |Ours| 0.16B|75.2|80.0|60.7|94.4|77.6 (+50.1)|
>
> From the results, SFT exhibits superior accuracy compared to PEFT methods (LoRA and DoRA), which can be attributed to its more trainable parameters, achieving an average accuracy 10.5% higher than LoRA. Notably, DoRA introduces an advanced scheme by decomposing pretrained weight matrices into magnitude vectors (m) and directional matrices (V), where LoRA is applied specifically to V while m is trained separately. This architectural innovation allows DoRA to surpass LoRA slightly in accuracy. Crucially, our proposed method achieves statistically significant performance improvements through two key innovations: 1) a more rational capacity decomposition strategy, and 2) a novel Mixture-of-Roles framework enabling dynamic interaction between decomposed modules. These enhancements collectively contribute to our method's marked accuracy superiority over SFT and PEFT methods.
>
> **2. Validating on more agent structures, such as Reflection, AutoGen.**
>
> Thanks for the valuable suggestion. Our method is not constrained by agent architectures. For example, the ​framework of Reflection[1] comprises three modules: ​Actor, ​Self-Reflection, and ​Evaluator. Similarly, in ​AutoGen's[2] application scenarios (Figure 3), multi-agent coding involves modules like ​Commander, ​Writer, and ​Safeguard. While each module in these frameworks operates as an independent LLM, our approach integrates multiple independent LLMs into a ​parameter-efficient Mixture-of-Roles architecture.
>
> Due to the limited time during rebuttal, directly extending our method to these application scenarios proved challenging. Therefore, we modified the agent architecture by adding or removing specific roles based on the existing ​BFCL multi-roles training dataset. The results are as follows.
>
> |Archs|Trainable Params|AST (Non-live)|Exec (Non-live)|AST (Live)|Relevance (Live)|AVG|
> |:-:|:-:|:-:|:-:|:-:|:-:|:-:|
> |Base|-|21.9|19.2|29.8|38.9|27.5|
> |2-roles|0.11B|60.6|66.3|48.3|79.5|63.7 (+36.2)|
> |3-roles|0.16B|75.2|80.0|60.7|94.4|77.6 (+50.1)|
> |4-roles|0.21B|70.8|74.2|57.6|90.3|73.2 (+45.7)|
>
> Our method integrates three core roles: Reasoner, Executor, and Summarizer. Through architectural modifications — specifically removing the Summarizer and introducing a Planner — we extend the framework to 2-roles and 4-roles. Experimental results demonstrate accuracy improvements across all variants compared to the baseline model. Notably, the 3-roles architecture achieves peak performance (accuracy improvement of 50.1% over baseline), whereas the 4-roles, despite containing more trainable parameters, shows diminishing returns (45.7% accuracy gain). This empirical evidence highlights that ​rational role definition and architecture design outweigh mere parameter quantity expansion in achieving optimal model performance.
>
> [1] Reflexion: Language Agents with Verbal Reinforcement Learning
>
> [2] AutoGen: Enabling Next-Gen LLM Applications via Multi-Agent Conversation
>
> **3. Validating on more agent-related datasets. How about generalization ability on other agent tasks, maybe refer to AgentGym.**
>
> Thanks. In our experiments, we conducted evaluations across multiple datasets including StableToolBench, BFCL, GSM8K, and MATH. In fact, the application scenarios of these datasets involve API calls, Java programming, Python programming, JavaScript programming, and mathematical problem-solving, etc., not just tool use. AgentGym is an impressive work, however, limited by the time of rebuttal, it is difficult for us to reproduce AgentGym in a short time and migrate our methods to it. However, we supplement the experiments of Llama3.2-1B-Instruct on BFCL with varying amounts of training data.
>
> |Train Data|AST (Non-live)|Exec (Non-live)|AST (Live)|Relevance (Live)|AVG|
> |:-:|:-:|:-:|:-:|:-:|:-:|
> |0|21.9|19.2|29.8|38.9|27.5|
> |1k |49.5|44.6|46.3|79.1|54.9 (+27.4)|
> |5k|55.3|51.8|50.6|82.1|60.0 (+32.5)|
> |10k|58.8|57.4|52.7|85.9|63.7 (+36.2)|
> |50k|70.4|74.9|56.5|91.7|73.4 (+45.9)|
> |90k|75.2|80.0|60.7|94.4|77.6 (+50.1)|
>
> From the results, even with only 1k training samples, we still achieves a 27.4% improvement in average accuracy, demonstrating its strong generalization capability. As the training data volume increases, the accuracy further improves accordingly.

---

### Decision · Program_Chairs · 2025-05-01

**Decision:**

Accept (poster)

**Comment:**

This paper present a parameter-efficient fine-tuning method (mixture of roles framework) for agent tasks and received consistently positive feedback from all four reviewers, i.e., three Weak Accept, one Strong Accept. The authors provided additional experiments and clarifications during the rebuttal to address the raised concerns, supporting a recommendation for acceptance.